# Nexus between Climate Change, Displacement and Conflict: Afghanistan Case

**Andrej Přívara [1],\* and Magdaléna Přívarová [2],\***

[1]   Department of Social Development and Labour, Faculty of National Economy, University of Economics in Bratislava, Dolnozemská cesta 1, 852 35 Bratislava, Slovak Republic

[2]   Department of Economics, Faculty of National Economy, University of Economics in Bratislava, Dolnozemská cesta 1, 852 35 Bratislava, Slovak Republic

\*   Correspondence: andrej.privara@euba.sk (A.P.); magdalena.privarova@euba.sk (M.P.)

**Abstract:** The character, the motion and the proportion of environment-induced migration have radically changed in recent years. Environment-induced migration is an increasingly recognized fact and has become one of the main challenges of the 21st century, and needs to be focused on to ensure sustainable growth. This new stance is due to the changing character of environmental degradation. Global environmental issues, including climate change, loss of biodiversity, river and oceanic contamination, land degradation, drought, and the destruction of rainforests, are progressively stressing the earth's ecosystems. Among these issues, climate change is one of the most severe threats. Climate change alone does not directly induce people to move but it generates harmful environmental effects and worsens present vulnerabilities. The current study aims to provide cornerstone links between the effects of climate change, migration decisions, displacement risk and conflicts in the example of Afghanistan, as a country that is extremely affected by both climate change and conflicts, and outline priority policy focuses to mitigate the current situation in the country.

**Keywords:** environment-induced migration; sudden-onset events; slow-onset events; displacement risk; climate change; vulnerability; conflicts

## 1. Introduction

The relationship between migration and the environment is not a recent phenomenon. Environmental considerations have always affected population mobility. Populations have often had a tendency to move in response to environmental changes. However, the character, the dynamics and the proportion of environment-induced migration have radically changed in recent years. Environment-induced migration has been an increasingly recognized fact and has become one of the main challenges of the 21st century, and needs to be focused on to ensure sustainable growth [1–7]. This new stance is due to the changing character of environmental degradation. Global environmental issues, including climate change, loss of biodiversity, river and oceanic contamination, land degradation, drought, and the destruction of rainforests, are progressively stressing the earth's ecosystems [8,9]. Among these issues, climate change is one of the most severe threats. The damaged equilibrium of the world's ecosystems is anticipated to have an increasing effect on the human environment, causing an extensive socio-economic vulnerability. In these conditions, climate change contributes to a new difficulty in the links between human migration and environmental degradation. Climate change alone does not directly induce people to move but it generates harmful environmental effects and worsens present vulnerabilities that make it challenging for people to survive in the territories where they live [10].

Climate change is manifested in sudden- and slow-onset events, which to a great extent increase displacement risk in the regions. Furthermore, slow-onset events accompanied by economic, social and political stress induce the long-term vulnerability of the regions.

An increasing number of studies are linking climate change to violence or conflicts; however, strong scientific evidence of this association remains unclear and disputed. The global data on international migration connected to environmental disasters and conflicts is limited. The data on how climate change events affect migration are particularly rare because it is difficult to insulate environmental factors from others [9,11–13]. Furthermore, identifying strong causal effects is likely to be complicated. Some studies have indicated correlations between rainfall and temperature change and conflict [14–18]. However, correlation does not always equal causation. While exploring associations between environmental change, migration and conflict in Africa, Freeman [19] concluded that the causation between environmental change and migration or conflict is indirect. Thus, a lack of proper data and the indirect nature of interlinks between climate change, migration and conflicts make it methodically difficult to assess interactions between them by employing econometric assessment methods. Relevant studies on the interaction between climate change, migration and conflicts in the affected areas are often based on extensive literature reviews and case studies. In particular, when investigating the climate change, migration and conflict interconnection, Burrows and Kinney [20] conducted a literature review on the associations between climate change and migration, and those between conflict and migration. The authors indicated that "climate change has the potential to lead to increased migration and increased risk of conflict" and the intricacy of the systems linking migration, climate and conflict, as well as the extent to which this system depends on economic, demographic and political drivers of a specific location. Based on literature reviews and case studies, Brzoska and Fröhlich [21] analyzed different pathways from environmental change to migration and conflict and concluded that "in some cases, environmental migration leads to violent conflict, in others the situation remains calm, or migration may contribute to the lessening of tensions". Reuveny [22] found "severe environmental problems play a role in causing migration, which, at times, leads to conflict in receiving areas". Freeman also emphasizes that particular attention should be paid to local-level impacts of environmental change, symptoms of conflict and adaptive forms of migration [19].

The research design of the current paper is the following. First, we outline key definitions in the area of environment-induced migration and consider the problem of assessing environment-induced migration. Then, based on available global data on environment-induced migration, we consider displacement risk and vulnerability to sudden- and slow-onset events. Then, we consider urbanization effects in relation to climate change trends. We present findings from the literature on climate change–conflict links. Further, based on data of both global and country-specific reports, we analyze climate change, migration and conflict links in Afghanistan, a country that has been suffering from severe climate change and civil conflict for a long time.

The contribution of the current study is three-fold. First, we provide evidence of diverse associations of urbanization with displacement risk. In particular, we provide evidence of the two-fold nature of urbanization in relation to its contribution to displacement risk. Second, we contribute to the literature on how climate change is linked to migration and conflict on the example of Afghanistan. Third, we discuss policy priorities to mitigate severe life conditions in the country.

## 2. Environment-Induced Migration Literature Review

### 2.1. Definition of Environment-Induced Migrants

The complexity of the environment-induced migration phenomenon is replicated in the number of terms used to define it: "climate refugee" [23] or "environmental refugee" [24–27], "environmentally displaced person" [28], "forced migration due to environmental change" [29], "environment-driven migration" [11] and "environmentally-motivated migrants" [30].

There is no internationally established definition for individuals migrating for environmental reasons; they have no legal basis and status that confers responsibilities on states. In fact, people induced to migrate due to environmental factors do not fall completely into the categories provided by the current international legal framework and demonstrate the restrictions of the current paradigm where migration is basically framed. The international studies emphasize that environment-induced migration is out of the scope of international protection but in great need in terms of assistance and defense [31,32]. There is increasing recognition that current gaps need to be bridged. In reality, countries are left to determine their own standards, rules and interpretations of present norms and human rights responsibilities and they propose different levels of protection. Currently, international initiatives on the protection of environment-induced migrants are being discussed giving rise to a rich but fragmented practice.

The following categories of migrants are usually considered as environment-induced migrants: those forced to migrate to avoid the worsening slow-onset deterioration of the environment; those forced to migrate more permanently due to recurrent events; those displaced by climate-related disasters, who often move temporarily; and those who "choose" to move as an adaptation strategy, in response to environmental pressures and other factors [33].

The International Organization for Migration defined environmental migrants as "people or groups of people who, for compelling reasons of sudden or progressive change in the environment that adversely affects their lives or living conditions, are obliged to leave their habitual homes, or choose to do so, either temporarily or permanently, and who move either within their country or abroad" [3]. In the next section, we consider the types of environment-induced mobility and issues related to its assessment.

## 2.2. Environment-Induced Mobility: Types and Issues of Assessing

Climate change is anticipated to make the environment hotter, increase peak precipitation and cause more extreme floods, storms and droughts. These changes, in turn, are expected to cause further population mobility. According to the United Nation International Strategy for Disaster Reduction (ISDR) [34]—droughts, storms and floods have enlarged three-fold during the past 30 years and there is a relationship between migration and climate change. Continuous environmental deterioration affects the present situation and causes different types of human mobility, in particular, temporary and circular migration (for instance, because of periods of drought), or permanent migration (due to desertification or the rise in sea level). These environmental hazards induce people to migrate internally or globally. The current studies specify that a great portion of environment-induced migration is taking place within national borders. Nevertheless, there is also evidence of a growing number of environment-induced migration across international borders [11,35–37].

Among the issues that affect the assessment of the expansion and the character of environment-induced migration is a lack of proper data [38]. In particular, global data on international movement in the context of environmental disasters is limited. Only a few remarkable cases have been examined so far, in particular, the Nansen Initiative, 2015, which "intended to identify effective practices and build consensus on key principles and elements to address the protection and assistance needs of people displaced across borders in the context of disasters, including the adverse effects of climate change" [39] and research conducted by Dina Ionesco, Daria Mokhnacheva and François Gemenne in 2017. The International Organization for Migration—Atlas of Environmental Migration—"clarifies terminology and concepts, draws a typology of migration related to environment and climate change, describes the multiple factors at play, explains the challenges, and highlights the opportunities related to this phenomenon" [40]. The official statistics on humanitarian visas by countries such as the United States (US), Brazil and Argentina in some cases can be also analyzed.

The fast-changing status of migrants prone to national conflicts also contributes to difficulties in identifying environment-induced migration flows [41]. Recent studies indicate that today's internally displaced people (IDPs) can become tomorrow's asylum seekers or refugees [42]. IDPs are those

individuals who have not yet crossed an international border. For instance, approximately 55% of the Afghan refugees and 85% of the Syrian refugees interviewed in Greece in 2016 reported that they had been IDPs before passing over the national border. An additional issue that makes the link between migration and the environment problematic to indicate is the difficulty of separating environmental drivers from economic political and social drivers of migration because environmental determinants are usually associated with economic social and political factors [43–46].

It is also often difficult to distinguish between forced and voluntary environment-induced migration [47–50]. Categorizing environmental migration as forced may be rather indisputable in cases of an environmental disaster. We can state that, at the early and intermediate stages of the environmental degradation processes, migration movements are more likely to be voluntary and to be experienced by the affected inhabitants as an act of adaptation. However, this distinction is not precise. There is a complicated chain of causality between environmental changes, the loss of economic opportunities and migration. Furthermore, the decision to migrate has to be considered in the context of possible options. The ability to migrate is a function of both social and financial capabilities and the most vulnerable people often cannot afford to migrate. Thus, taking into account the complication of the associations between environmental change and migration movements, recent estimates of the number of people migrating in response to environmental changes, either directly or indirectly, either permanently or temporarily, either within their countries or cross borders, vary considerably [10,51,52]. In the next section, we discuss how displacement risk appears in terms of sudden- and slow-onset events.

## 3. Displacement Risk and Vulnerability to Sudden- and Slow-Onset Events

### 3.1. Sudden- and Slow-Onset Events: Contribution to Displacement Risk

Climate change events are being considered from the point of sudden- and slow-onset events [47]. According to the United Nations Refugee Agency (UNHCR) [53], sudden-onset events comprise "meteorological hazards including tropical cyclones, typhoons, hurricanes, tornadoes, blizzards; hydrological hazards including coastal floods, mudflows; or geophysical hazards including earthquakes, tsunamis, volcanic eruptions." Slow-onset events include "sea level rise, increasing temperatures, ocean acidification, glacial retreat and related impacts, salinization, land and forest degradation, loss of biodiversity and desertification".

Disaster displacement refers to "situations where people are forced to leave their homes or places of habitual residence as a result of a disaster or in order to avoid the impact of an immediate and foreseeable natural hazard. Such displacement results from the fact that the affected people are exposed to a natural hazard in a situation where they are too vulnerable and lack the resilience to withstand the impacts of that hazard" [53]. It should be noted that disaster displacement by its nature differs from other related notions of "migration" and "planned relocation." Migration refers to voluntary human movements and in the context of slow-onset natural hazards, "environmental degradation and the long-term impacts of climate change, such migration is often used to cope with, "avoid or adjust to" deteriorating environmental conditions that could otherwise result in a humanitarian crisis and displacement in the future". Planned relocation, in turn, refers to "a planned process in which people or groups of people move or are assisted to move away from their homes or places of temporary residence, are settled in a new location, and provided with the conditions for rebuilding their lives." Planned relocation can be voluntary or involuntary, and often occurs inside the country, but may, in remarkable cases, also be cross border [53].

Displacement due to natural disasters is more often associated with sudden-onset events and its association with slow-onset events is often dependent on whether the slow-onset event has turned into a natural disaster situation that affects individuals with no other rational option than to move [54].

There are important links between sudden-onset and slow-onset events. For instance, drought is a severe natural event. At the same time, it is also closely related to slow-onset events and accumulative

climatic change [55]. The interaction between sudden-onset and slow-onset events may result in ecological thresholds being crossed [56]. Ecological thresholds happen when external influences, positive reactions, or nonlinear uncertainties in a system cause changes to spread in a domino-like manner that is potentially irreparable. If an ecological threshold is crossed, the environment is not likely to come back to its former state.

Due to the long-term nature of slow-onset events, people typically a choice have to some extent to decide when they migrate, or in other words, when they accept that the situation has become a disaster that leaves them with no other alternatives than to move. The ecological threshold is explicit when the slow-onset event becomes a disaster generated by a sudden-onset. For instance, in the case of the sudden loss of housing due to flooding induced by sea level rises.

However, when the disaster is caused by a severe interruption of livelihood induced by a slow-onset event, for instance, in the case of food insecurity caused by desertification, the decision to migrate predictably implies a subjective perception of the risks associated with the slow-onset event-generated disaster. In the mentioned cases, any decision to migrate would have perceptions of the risks of leaving and perceptions of the risks of staying as vital determinants for which perception of risk is subjective [57,58].

Slow-onset events may turn into a sudden disaster and contribute to displacement risks in four key ways: (1) through diminished ecosystem services, including access to basic human needs such as fresh water, food, energy production, shelter, and those that are necessary for human beings. The lack of vital resources may cause the severe turmoil of livelihoods. In case this turmoil of livelihoods crushes the community's capability to handle the hazards, the situation becomes a disaster and displacement risks become more serious. Slow-onset hazards may, for instance, in combination with other factors, cause critical food insecurity as their effects on food production affect naturally based livelihoods, because they are based on agriculture, fisheries, hunter gathering, pastoralism or horticulture livelihoods. When communities are not able to handle severe food insecurity, they can be displaced to survive in other locations that suggest food security [59,60]; (2) by becoming a disaster provoked by the sudden-onset event. Numerous slow-onset events effects are in fact sudden-onset events. For instance, when the sea level rises quickly and this turns into flooding, desertification becomes a wildfire, or temperature rises turn into heat waves. In these cases, sudden-onset events may exacerbate the risk of displacement [39]; (3) by deteriorating community's and ecosystem's readiness to resist the impacts of sudden- and slow-onset events if caused by a cascade of hazards, provoking displacement. If livelihoods do not recover after a disaster, either prompted by a slow or a sudden-onset event, even if less stressful, this can push households into a situation of critical humanitarian need; (4) furthermore, slow-onset events are frequently a hidden provoking factor in many contexts, operating as a hazard multiplier for other determinants of crisis such as economic, social, political and cultural factors [39].

These factors, as mentioned above, have become difficult to distinguish one from another, and may result in humanitarian crises, inducing internal and cross border migration and displacement. Sudden- and slow-onset events, even though they are not a direct cause of violent conflict, can worsen already fragile conditions. They may provoke conflict over resource shortages and are often considered as a multiplier of preceding clashes.

It can be argued displacement is the outcome of the mixture of exposure, vulnerability and the intensity of the slow-onset events. Societies that are characterized by the same level of exposure due to their specific geographical position in hazard-prone areas and their exposure to geological and hydro-meteorological hazards and climate change may be characterized by diverse vulnerabilities specific to their economic, social and environmental challenges [61–67]. Vulnerable people have the smallest opportunities to adapt locally or to migrate away from risk and, while moving, often behave this way as a last possibility. In the next section, we analyze the exposure of the regions to sudden-onset events.

*3.2. The Link between Slow-Onset Events and the Vulnerability of Migrant Populations*

Sudden-onset events can bring great risk of displacement for the population due to quick, extreme and often unpredictable occurrence and harm. However, the negative effect of slow-onset events as mentioned above maybe two-fold and harm communities in both the short and long term. Thus, if the region is prone to slow-onset hazards, its vulnerability to climate change is initially long-term and requires, thus, more productive and intensive measures to cope with the adverse effects of slow-onset events, focusing on maintaining the established structure of the economy.

Vulnerability is directly associated with an adaptive capacity to climate change as increased vulnerability signalizes that individuals are likely to have a decreased adaptive capacity—in other words an ability to respond to the effects of climate change. Adaptive capacity in turn affects the ability to migrate and the freedom of choosing to behave this way, which in turn increases their vulnerability during and after movement. The vulnerability of migrants is frequently generated or exacerbated by intensified barriers to international migration, which exacerbates criminalization; border restrictions; a lack of regular migration flows, including for work purposes, family unity, education, and humanitarian needs; migration policies based on deterrence [68].

Vulnerability increases because of complex interactions with forms of inequality, discrimination, both societal and structural, that weaken and create the uneven enjoyment of rights and levels of power [49]. However, the adverse impacts of slow-onset hazards can aggravate present inequities. Slow-onset hazards can cut off and considerably limit the access to medical services and education, especially in underserviced areas. Women and children in vulnerable situations face the greatest and disproportionate risks from the negative effects of climate change, which can expand gender inequalities [69,70]. These women face the inequality of accessing the human social freedoms and means necessary for the realization of social, economic, and cultural rights, which can cause losses in times of environmental pressure. The indigenous population faces greater risks from slow-onset hazards as they negatively affect their self-determination, livelihoods, resources, territories, rights and culture. Slow-onset hazards disproportionately harm the indigenous population relying directly on its environment to meet its necessities, threatening, thus, the effective use of rights to water, food, and health.

The social, economic and political drivers of displacement inevitably require effective measures for increasing the adaptive capacity of the regions. However, a growing tendency of urbanization in many regions of the world brings additional displacement risks, associated not only with competitions for resources in urban areas, but also with specific consequences of urbanization, which we will discuss in the next section in more detail.

## 4. Urbanization and Displacement Risks

Rapid urbanization is increasing the vulnerability of regions to the effects of climate change, according to a new United Nations report [71] and increasing displacement risks. More than 50% the world's population live in cities and approximately 2.5 billion people are projected to join them by 2050. Torrential rain and storm surges are becoming more frequent in big and densely populated cities like New York and Mumbai, hitting the population living in informal settlements like slums the hardest. Over 90% of this growth will occur in Latin America, Asia, Africa and the Caribbean. The report highlights that the features that make their citizens especially vulnerable to the negative effects of climate change globally, particularly in developing countries, often characterize urban centers. Desertification damages arable land needed to feed urban inhabitants, sea levels rise, as mentioned above, which threatens everyone living in small-island countries, coastal areas and delta regions. Rapid urbanization, thus, is considered as a factor that is making society and economy increasingly vulnerable to the effects of climate change.

Climate change brings a number of specific hazards not only to vulnerable geographic territories but also to destination cities. The more affected people move to safer cities, the higher the impact on destination cities. The changing temperature problem in destination cities has already been

recognized [72–76]. The temperature in cities can be several degrees higher than the temperature in neighboring areas. Urbanization, including human activity development, changes in land use, developments in heat emissions, and dense building has a huge effect on a city's local climate. One of the effects of urbanization is the so-called urban heat island (UHI) effect, which occurs when urban cooling rates are slower than rural ones. The main factors that may provoke the temperature difference between urban and rural territories are the following: dense development in urban territories, which diminishes the speeds of the wind and constrains convection cooling; high-density buildings in urban territories limit the view of the sky and diminish the heat release back to space; high building heat capacity in the city compared to neighboring rural territories, resulting in more of the energy of the sun being absorbed and accumulated in the city; man-made heat emissions by buildings, transportation and industries and air conditioning in the cities.

Rapid urbanization during 1979–2010 likely contributed to a prudent warm bias in unhomogenized world changes in land temperatures. During the above-mentioned period, urban stations warmed approximately 10% faster than rural stations. The authors found that urban locations warmed quicker than rural ones across all cutoffs, urbanity proxies and spatial resolutions surveyed [77].

Su et al. researched the impact of urbanization on local weather in Northern China (Jing-Jin-Ji District) [78]. The authors used the Weather Research and Forecasting (WRF) model in order to quantitatively investigate the effects of previous urbanization and potential possible prospective urbanization on the regional weather in Jing-Jin-Ji District center. The research was based on hydrometeorological data from two weeks in July 2019. The authors found that the main impact of urbanization on climate resulted in the effects on the maximum temperature and peak precipitation. In particular, urbanization in the Jing-Jin-Ji district contributed to the increased daytime and night temperature, as well as the temperature difference between night and day, while urbanization decreased the total rainfall and peak precipitation.

Urban areas in North America, South America, Europe, the Middle East, Asia and Australia [79] experienced more hazardous heat stress nights than rural territories as a result of a higher current day heat stress during 2000–2016. In Korea, a significant positive correlation between an increase in temperature and local population growth has also been reported, which indicated that urbanization had a significant contribution to the temperature increase in 40 Korean cities during 1975–2005 [80–82]. In the next section, we consider findings from the literature on how climate change is linked to conflicts.

## 5. Climate Change–Conflict Links

Climate change is hypothesized to trigger or aggravate conflicts through sudden- and slow-onset events as they threaten livelihoods, trigger poorly designed climate action with unintended consequences, strengthen cleavages, increase competition, diminish state capability and legitimacy, and lead to large migration that may negatively affect host areas [16]. Some quantitative studies found substantial statistical correlations between climate change and violence or conflict and identified that if communities responses to climate change remain unchanged, climate change has the potential to exacerbate conflict and violence [83–85]. However, as mentioned above, it is difficult to determine the direct impact of climate change on migration and conflicts. Thus, relevant studies often largely criticize large-scale quantitative works on exploring these links, in particular regarding statistical methods, sample selection and lack of explanation of causal links [86,87]. According to recent research by SIDA, "there is no direct and linear relationship between climate change and violent conflict, but under certain circumstances climate-related change can influence factors that lead to or exacerbate conflict" [88].

Nordqvist and Krampe [89] analyzed literature on climate-conflict links in South Asia and South East Asia. The authors indicated that climate change can induce violent conflict in the territory in the following cases: (1) it leads to a worsening in people's livelihoods; (2) it affects the tactical reflections of armed groups; (3) elites use it to exploit social resources and vulnerabilities; (4) it leads to displacements and increases levels of migration.

Van Baalen and Mobjörk [90] researched the links between climate change and dynamics of conflict in East Africa. They detected four key mechanisms in East Africa: (1) deteriorating livelihoods; (2) increased migration and changes in pastoralist migration; (3) elite detention of local dissatisfaction; (4) strategic reflections among armed groups.

Rüttinger et al. [91] distinguished seven compound climate-conflict risks that emerge when climate change interacts with other economic, social and environmental tensions, which are not isolated from each other but interact in complex ways. The seven climate-fragility risks detected are:

(1) local resource competition: given the pressure on natural resources rises, competition can cause instability and conflict in conditions of a lack of effective management;

(2) unplanned results of climate policies: given climate adaptation policies are more widely executed, the risks of unintentional negative effects will also increase;

(3) transboundary water management: transboundary waters are often a source of pressure; as demand grows, competition for water use will increase the pressure on present governance bodies;

(4) extreme weather events and disasters: disasters will worsen vulnerability challenges for populations, especially in conflict situations;

(5) volatile food prices: the effects of climate change are likely to disturb food production in many regions, increasing market volatility and prices, and enhancing the risk of disapproval and civil conflict;

(6) livelihood insecurity and migration: the effects of climate changes will worsen the insecurity of those communities which depend on natural resources, which can force them to migrate or become engaged in illegal hustles;

(7) sea-level increase and coastal degradation: an increase in the sea level will threaten the feasibility of low-lying areas before they are flooded, which will cause displacement, social disruption and migration, and inequality in relation to maritime boundaries and ocean resources may rise.

Freeman, while exploring the links between environmental change, migration, and conflict, proposed five scenarios characterizing the links between them (Table 1) [19].

**Table 1.** Pathways connecting environmental change, migration, and conflict.

| Scenarios | Links between Environmental Change, Migration, and Conflict |
|---|---|
| Scenario 1: Abundance | Environmental change → migration → conflict |
| Scenario 2: Scarcity | Environmental change → constrained migration → conflict |
| Scenario 3: Conflict-induced migration | Conflict → migration → environmental degradation → conflict |
| Scenario 4: Environmental degradation as a method of conflict | Conflict → environmental degradation → (constrained) migration |
| Scenario 5: Independently occurring climate change and migration lead to conflict | Climate change + migration → conflict |

In the next section, we discuss climate the change situation in Afghanistan as well as consider the links between climate change, migration, and conflict in the country.

## 6. Climate Change, Migration and Conflict Links in Afghanistan

### 6.1. Climate Change in Afghanistan

Climate change in Afghanistan is mainly associated with temperature rises. Afghanistan has experienced a temperature rise considerably higher than the mean worldwide, amounting to 1.8 °C between 1951 and 2010. The temperature rise is anticipated to last from 2006 until 2050 and rise by 1.7–2.3 °C and afterwards by 2.7–6.4 °C until 2099 through the entire country [92].

A detailed analysis of climate change in Afghanistan is limited due to mentioned above lack of data, also including climatic data. However, striking droughts in Afghanistan have become a solid feature of its climate. Several severe droughts have been recorded with a tendency to increase the

frequency of the drought cycle, for instance, 1963–64, 1966–67, 1970–72 and 1998–2006. The period 1998–2006 appeared to be the longest and most extreme drought in the climate history of Afghanistan. Over the last 3 years, most regions in Afghanistan have experienced between 4 and 6 successive rainfall seasons that were far below the average rainfall previously. As a result, there have been recorded substantial decreases in water tables, river flows, snow depths, water levels in dams and soil damp. According to the Famine Early Warning System (FEWS), in July 2018, in 22 out of 34 provinces of Afghanistan, its cumulative rain and snowfall during October 2017 to May 2018 was 30% to 60% below average [93]. These climate change events have already negatively and irrevocably affected agricultural production in Afghanistan. Repeated cycles of drought caused a loss of crops, livestock and livelihoods and weakened purchasing power. According to Integrated Food Security Phase Classification (IPC) [94], as of September 2018, 9.8 million people (43.6% of the rural population) were recorded to be in Food Crisis and Emergency, indicating that Afghanistan is experiencing a major food and livelihood crisis, which has been mainly caused by the severe drought restricting food production and diminishing the livelihoods and assets of farmers and other livestock keepers.

As of 8 July 2019, the impact of drought has severely influenced population migration within Afghanistan and caused the displacement of 287,000 people due to drought, primarily from the north-western and western regions. The sudden inflow of over a quarter million people into the borders of Herat, a provincial capital city, during just a few months, has caused the appearance of 19 massive and expansive informal settlements. According to estimations, 13.5 million people are severely food insecure and need emergency assistance. In these conditions, displaced households residing in impermanent and poorly protected shelters are facing the risk of severe winters and the high risk of flooding, especially those living on dry-river beds [95].

Furthermore, currently all the regions of Afghanistan are suffering from floods (Table 2). El Nino conditions, stated in February 2019, brought above-normal snowfall/rainfall to Afghanistan and higher temperatures across the country. This was accompanied by poor soil absorption and limited vegetation in many areas as a result of drought, and the massive rains in March and April appeared to be a catalyst for floods in spring. Rainfall analysis for the year 2019 specifies higher than average rainfall, higher temperatures that caused floods and landslides and, consequently, the jam of traffic-intensive areas during these months. The issue of flash flooding was exacerbated by the narrow valleys, which triggered floodwaters to pass through villages, abolishing many houses.

**Table 2.** Sudden- and slow-onset disasters in Afghanistan by regions.

| Region | Province | Disaster | Districs |
|---|---|---|---|
| East Region | Nangahar | Flood | Hesarak, Sorkh-Rod, Aftkamena, Batikot, Haychen, Kama, Khewa, Daray Noor, Chaparhar, Baesot and Moman dara |
| | Kunhar | Flood | Sar-Kani, Marwara, Shegal, Sheltan, Watapor, Dangam, Asmar, Narahy, Ghaziabad and Chapa Dara |
| West Region | Farha | Flood | Nawahe Shar, Bakwa, Balabolok, Khak Safed, Anar Dara, Sheb Ko, Posht-Rod and Posht-Ko |
| | Herat | Flood and Drought | Herat city, Zer-ko shendan, Posht ko shendan Zawol, Khush ke Kuhna, Gulran |
| | Nimroz | Flood and Drought | Chakhansour, Kang, Char Burjak, Khashrood, Damarda and Mirza Azim |
| | Badghis | Flood and Drought | Abkamry, Moqure, Bala Morghab, Jawand, Nawe - Laman and Badghis city |
| South West | Helmand | Flood | Nawa, Lashkargah, Nada Ali and Nahri Saraj Malgir |
| | Kandahar | Flood and Drought | Kandahar city, Dand, Maiwand, Panjwai, Spin Boldak, Zheri, Arghandab, Daman, Takhta Pul and Arghestan |
| North Region | Jowzjan | Flood and Drought | Khanaqa, Khwaja Dako, Qush tepa, Faizabad and Shiberghan city |
| | Sar-e-Pul | Flood and Drought | Suzma qala, Ghusfandi, Sayad, and Capital of Sare-e-Pul |
| | Faryab | Flood and Drought | Shirin Tagaab, Qesar, Pashtoon Kot, Dawlat Abaad, Almar, and Khuja Sabz Posh |
| | Balkh | Flood and Drought | Chamtall, Dehdadi, Sholgara, Doko and Part of Balkh City |
| North West Central Region | Kabul | Flood | Districts: 17th, 11th, 19th, 9th, 5th, 13th, 18th and Qarabagh, Shakar Dara and Surabi Districts of Kabul City |

Source: International Federation of Red Cross and Red Crescent Societies [96].

Floodwaters also damaged farmlands and irrigation facilities. Afghanistan communities lack many support services, which makes them more vulnerable to impacts from floods. The floodwaters damaged equipment, infrastructure and facilities, which resulted in disruption of the routine functioning of services in the affected territories. Several of the public buildings, which were spoiled by the disaster, became unreachable due to the destruction of roads and bridges. It should be noted that the negative impact of drought and floods is exacerbated by the ineffective use of natural resources in the country. Nevertheless, Afghanistan has broad water resources that, according to the National Environmental Protection Agency of Afghanistan (NEPA), amount to nearly 1700 m3 per capita per year, which is theoretically an adequate amount for domestic environmental, industrial, agricultural use. However, the country still does not use natural resources properly due to a lack of water management efficiency because of the huge annual variability in water availability. Despite Afghanistan being strongly dependent on agriculture, a sector which employs approximately 85% of the population and contributes nearly 30% to the GDP, there is a lack of estimations and projections on the future trends and consequences of such dependency [97]. The negative effects of climate change are reflected not only in the agricultural sector but also in other sectors, indicating broad negative impacts (Table 3).

**Table 3.** Key climate change impacts in Afghanistan.

| Climate Change Stressors | Risks |
|---|---|
| **Agriculture** | |
| Temperature increase; Drought; Changes in rainfall patterns and snowmelt | Decreased soil liquid availability during planting |
| | Less continual rain during peak cultivation season, causing yield decreases |
| | Crop decrease due to lack of water |
| | Livestock migration, famine and/or forced sale |
| | Reduced availability of animal feed |
| **Water resources** | |
| Reduction in glaciers and snow cover; More rapid and earlier spring snowmelt; Drought | Increased risk of flash flooding, intensified by the influence of drought (increased soil impermeability) and land degradation |
| | Diminished river flows |
| | Lack of irrigation resources |
| | Reduced water supply and hydropower potential |
| **Human health** | |
| Increased temperatures; Drought; Changes in precipitation patterns | Worsened food security, hunger and dependence on food aid |
| | Increased casualties due to natural disasters, such as droughts, flood-induced landslides and floods |
| | Extension of the altitudinal range of mosquito vectors, increasing the population at risk |
| | Increased frequency of waterborne illnesses such as cholera and diarrheal disease |
| **Ecosystems** | |
| Increased temperatures; Drought | Worsened ecosystem services including soil filtration and water quality |
| | Increased desertification |
| | Increased pressure on habitats for migratory birds (wetlands) |
| | Diminished snowpack and increased vulnerability of high mountain biodiversity |
| **Governance and conflict** | |
| Increased Temperatures; Drought; Extreme climatic events | Competition over the use of productive rangelands which are in shortage |
| | Increased internal displacement |
| | Exacerbated regional stresses or conflict |
| | Increased poppy (opium) production, a drought-resistant crop, despite poppy eradication efforts |

Source: USAID [98].

Afghanistan faces extremely negative and broad risks of further worsening its insecurity and instability. Furthermore, as mentioned above, Afghanistan has also been suffering from civil conflict for a long time, one which still continues now. This is discussed in more detail below.

### 6.2. Conflict and Migration Situation in Afghanistan

Migration patterns in Afghanistan are indicated by internal displacement and migration and massive emigration and refugee outflows since 1970s. There has been conflict in Afghanistan since the 1979 Soviet invasion and this is considered as the main reason of migration. The first huge migration occurred in 1979 after the Soviet invasion, when the population mainly migrated to Iran and Pakistan. Some refugees returned after Soviet withdrawal but in 1990, Mujahedeen (warlords) gained power and civil conflict ignited, causing the next huge migration. In 1995, the Taliban gained power until 2001. It should be noted that during this period, Afghanistan was hit by a severe drought, which appeared to be more severe, longer and devastating than the other droughts that had occurred in Afghanistan before.

The Karzai administration gained power in 2001 and North Atlantic Treaty Organization (NATO) coalition forces came to the country to defeat Al-Qaida and Taliban. This caused the next migration and then the presidential elections and the withdrawal of NATO coalition powers from Afghanistan caused another huge migration during 2014–2015 [99].

It should be noted that decades of conflict have trapped the population of Afghan in a persistent crisis. The continued conflict across huge areas of the country, including land action, inflight action, and the indiscriminate use of improvised explosive devices (IEDs), is triggering extremely dangerous levels of psychological and physical harm. Already during 2019, 98 suicide attacks have occurred. More organized violations of international humanitarian law (IHL) and international human rights law (IHRL) continue, including targeted killings, the forced recruitment of children, and attacks on health and education facilities. According to estimations, in 2019, approximately 250,000 Afghans will need emergency medical treatment because of continued conflict, and approximately 1.8 million people live within one kilometer of polluted areas with explosive hazards.

Current civil conflicts generate both an abrupt and long-lasting burden for the population, exposing them to sudden and distressing attacks and leaving them exposed to unexploded weaponry—both of which cause significant trauma-related problems. This situation, accompanied by a severe drought hit in 2019, has caused huge migration flows. Yet, in 2018, approximately 550,000 Afghans have been pushed to leave their dwellings either due to a loss of livelihood as a result of conflict or drought [100].

Both conflicts and climate change have negative impacts on the population and living conditions. However, our review of the literature helped to outline that when interacting, climate change may generate cumulative effects currently existing in conflicts, in particular, in the face of new conflicts, which we discuss in the next section.

### 6.3. Cumulative Effects of Conflict and Climate Change

#### 6.3.1. Conflict Associated with Restricted Access to Vital Services

One of the main negative cumulative effects of continuing conflicts and drought hits, including the current hit, is limited access to basic services, in particular, to health and education. According to estimations, in 2018, nearly 4000 h of health care delivery have been lost. Due to forced attacks against medical workers and services, which grow in both deadliness and frequency, and the forced closure and devastation of health facilities, approximately 335,000 consultations were missed. While restricted access to vital services affects all populations, returnees and internally displaced people are predominantly marginalized due to either their loss or lack of proper civil transcripts. Other development indicators also stay obstinately low or are declining. In particular, in two-thirds of Afghanistan's provinces, the frequency of global acute malnutrition (GAM) is above crisis thresholds; one-third of Afghan children are not immunized. Continuing conflict impedes parental and child

health, especially in rural areas, where approximately 75% of women live. According to forecasts, nearly 2.64 million people will need humanitarian assistance due to restricted access to services, including 1.9 million underfed children and nurturing mothers. Furthermore, approximately 4500 schools and other civilian infrastructure may also become exposed to forced attacks, violence or damage, and force parents to keep children away from school, necessitating preventive and defensive measures to be put in place [100].

### 6.3.2. Conflict Associated with a Lack of Regional Cooperation and Control over Water Sources

Water access is a regional critical point due to the fact that four of Afghanistan's main rivers are transboundary and future water demand can increase regional pressures [100]. There is a lack of regional cooperation and control over water sources due to political incitement in the past. Droughts and diminished river flow from prior snow melt will likely increase the stress on deficient water resources. Furthermore, non-State Armed Groups, especially the Taliban, are progressively making use of the entrenched regional strains for their own strategic benefits.

The projections of lower water accessibility make water-intensive principal crops such as wheat less appealing to farmers, who will likely choose less desirous crops, mainly opium poppy, which gives approximately three times the income per hectare [101]. Afghanistan remains the world's largest producer of opium and hashish, accounting for 90% of the global supply. It should be noted that drug production and trade in Afghanistan is associated with conflict and the security situation in the country. The opium trade provides enormous incomes to speculators and anti-government rebels, contributing to the status quo of poor governance [101]. By enhancing the attractiveness of poppy production, deteriorating climatic conditions are exacerbating insecurity and instability in Afghanistan.

### 6.3.3. Conflict Associated with a Lack of Regulated and Sustainable Land Management

Approximately 80% of population in Afghanistan is employed in farming. The combined negative effects of conflict, droughts, and a lack of regulated and sustainable land management have considerably contributed to Afghanistan desertification and land degradation. Increasing demand for land driven by fast population growth and the massive inflow of returning refugees exacerbates this trend. Linked to fast increasing environmental degradation accompanied by climate change, the likely result is increased competition for land in rural and urban areas. This in turn risks exacerbating pressures, activating complaints and inducing insecurity and instability [101]. Clashes over access to land are considered among the most frequent cause of conflicts and violence in the country [102]. One of the land-related conflicts is the deep-seated inter-ethnic clash between the inhabited Hazara and the nomadic Kuchi in the central highlands of Afghanistan that was caused by an overlap of legal historical rights held by the Hazara and legal rights held by Kuchi, who compete for access to the land due to a high dependency on high-altitude grazing land needed for livestock and restrict the access of the other resource-users [103]. The situation is also tense due to the return many refugees, as, since 2002, more than 5.8 million Afghan refugees have returned. The concerns around land rights and the lack of agreement on managing the refugee returnee situation has triggered local tensions toward incoming refugees [101].

### 6.3.4. Conflict Associated with Growing Urbanization

In addition to the above-mentioned tensions related to the return of refugees to rural areas, concerns are also growing with respect to their return to urban areas. Nevertheless, Afghanistan stays an agrarian country, and its urban areas are growing quickly. Due to the effects of economic development, sudden- and slow-onset disasters and conflict, the urban population is growing and returning refugees are greatly contributing to this growth. The lack of health, infrastructure for shelter, electricity provision and water makes urban areas overcrowded and vulnerable to poverty as people reside in unplanned and illegal settlements. This situation poses a challenge for the Afghan government bodies, exacerbating the profound economic and social instability facing the country. In the current

situation, the capacity of the government to insure sustainable employment prospects, especially to young men, can turn into source of irritation and a specific conflict driver [104].

## 7. Conclusions and Discussion

Environment-induced migrants are extremely vulnerable, first, due to the fact that they have no legal basis and status that confers responsibilities on states and do not fall completely into the categories provided by the current international legal framework. Among the issues of assessing real expansion and the character of environment-induced migration is a lack of proper data and the fast-changing status of migrants prone to national conflicts also contributes to difficulties in identifying environment-induced migration flows.

Displacement due to natural disasters is more often associated with sudden-onset events and its association with slow-onset events is often dependent on whether the slow-onset event has turned into a natural disaster situation that affects individuals with no other rational option than to move. The interaction between sudden-onset and slow-onset events may result in ecological thresholds being crossed.

Slow-onset events in particularly cause severe displacement risk and harm for the regions. First, this is because the ecological threshold is crossed and the ecosystem is not likely to return to its former state in this case. Second, slow-onset events have the potential to harm populations and areas and increase displacement risk in the long-run in ways in which sudden-onset events are unlikely to. Third, sudden-onset events may reflect only short-term threats to regions and, in the case of proper priorities in economy and policy development, their harmful effect can be managed to some extent. However, slow-onset events may include both short- and long-term vulnerabilities and, in some regions, pose much greater displacement risks.

Our research has shown that urbanization effects are also considerable. Urbanization appears to be one of the drivers of displacement in low-developed regions as it is often accompanied by the expansion of slums and other informal settlements, which are to a great extent vulnerable to sudden-onset events and significantly increased displacement risk. Thus, in this case, urbanization is a driver of displacement risk. At the same time, urbanization can also be perceived as an economic implication of displacement as the affected individuals often move from vulnerable rural areas to cities.

Furthermore, our analysis has shown that urbanization by itself may contribute to temperature increases in the cities by human activity development, changes in land use, developments in dense building, heat emissions, etc. and has a great effect on the city's local climate. One of the effects of urbanization is the so-called urban heat island (UHI) effect, which occurs when urban cooling rates are slower than rural ones. Thus, urban planning should be carefully developed as an element for the adaptive capacity stratagem of regions.

The literature review has identified that there is no direct and linear relationship between climate change and conflict or violence. However, climate change may trigger or aggravate conflicts through the sudden- and slow-onset events as they threaten livelihoods, trigger poorly designed climate action with unintended consequences, strengthen cleavages, increase competition, diminish state capability and legitimacy, and lead to large migration that may negatively affect host areas.

Our findings on the analysis of climate change, migration and conflict links in Afghanistan are the following. The country is extremely affected by drought and floods. The key climate change impacts in Afghanistan are observed from temperature increase, changes in rainfall patterns and snowmelt, a reduction in glaciers and snow cover, changes in precipitation patterns on agriculture, water resources, and human health.

Current civil conflicts generate both an abrupt and long-lasting burden for populations, exposing them to sudden and distressing attacks and leaving them exposed to unexploded weaponry—both of which cause significant trauma-related problems. This situation, accompanied by severe drought hit in 2019, has caused huge migration flows and conflict.

Both conflicts and climate change have a negative impact on the population and living conditions. However, our review of the literature helped to outline that when interacting, climate change and currently existing conflicts may generate cumulative effects, in particular, in the face of new conflicts, such as conflicts related to limited access to vital services, water-related conflict, land-related conflict and urbanization related conflict.

If considering scenarios proposed by Freeman for identifying links between climate change, migration and conflict, it can be implied that Scenario 2: "Scarcity" is the case of Afghanistan as migration in the country is caused by climate change in the face of a slow-onset events (drought) and a sudden-onset event (floods) and, as mentioned above, migration appeared to be a driver of other conflicts. However, it seems, Scenario 3: "Conflict-induced migration" can be also applicable in the case of Afghanistan as civil conflict was also the initial driver of great migration flows including those toward urban areas that enhanced pressure on natural resources and then fueled tensions. Such scenario considerations contribute to a better understanding of initial points that government policies should be focused on. However, we tend to conclude that the links between climate change, migration and conflict are heavily dependent on local conditions. In particular, in the case of Afghanistan, two initial drivers of migration have been identified, severe droughts and civil conflict, which, furthermore, when interacting, generate other conflicts. Thus, nevertheless, the resource system is suffering from both climate change and conflict. However, it represents only one of symptoms of a much broader political problem. In particular, measures to improve agricultural productivity will hardly contribute to instability and insecurity mitigation, as it is more vital to manage yield distribution and land rights over the already living population and returning refugees. Thus, we emphasize the importance of developing a legal system in the area of land rights.

As mentioned above, after many years of conflict and droughts, the present intensification of fighting and growing insecurity is impeding the Afghan population from accessing humanitarian aid and basic services, which should be the second policy priority. In the case of the further worsening of access to basic services, Afghanistan will be exposed to losing working capital that reflects an extremely dangerous trend. Currently, the Afghan population seeks to move to more favorable urban areas where employment and services are better, thus enhancing the negative effects of urbanization in the face of the uncontrolled growth of illegal slums that results in the increase in poverty levels, worsening health and overall vulnerability. Thus, cooperation between national and municipal authorities should be intensified toward the development of urban development projects aimed at improving living conditions and opportunities for populations living in slums as well as discussing immediate needs.

In this research, we have highlighted the lack of attention international law pays to the proper identification of environment-induced migrants. Thus, a fruitful area for future research is shaped by the need to determine the gaps in international and national laws in order to outline clearer international policy measures for coping with environment-induced migration issues.

**Author Contributions:** A.P. worked on the conceptualization and general orientation of the paper and took care of most of the bibliographical analysis and also analyzed the data. M.P. provided valuable insight, literature and text about sustainability aspects. All authors contributed to the writing of the article. All authors provided critical feedback and adjusting the structure of the work, analysis of results and revision of the manuscript according to the norms of the scientific journal.

**Funding:** This research was funded by Vega research project no. 1/0001/16: "Present and prospective changes in employment and related processes in the context of meeting the objectives of the European Employment Strategy" and VEGA research project no. 1/0287/19 "Integration of immigrants in EU countries from the point of view of migration policies".

**Conflicts of Interest:** The authors declare no conflict of interest.

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
