# Peer review of "Nexus between Climate Change, Displacement and Conflict: Afghanistan Case"

_sustainability, doi:10.3390/su11205586_

Round 1

Reviewer 1 Report

The paper does what its title says: It identifies key interactions between the different factors of environment-induced migration. But it certainly does not go beyond this: Neither is there a rigorous analysis of these interactions nor are there policy conclusions (at least none to speak of). Thus, the paper does hardly contain any original research. It gives more or less an overview of the problem and its dimensions. As such it may certainly be interesting to the general reader, hardly to experts in this field.

What I can say, as a reviewer, is that the presentation needs to be improved considerably - in particular with regard to language and style. Although comprehensible, the language is certainly not free from mistakes and the style is rather bad. E.g., "to indicate" (p. 3, l. 119) is probably supposed to mean "too intricate"; and the "ecological thresholds" are not "overlapped" (p. 4, ll. 166-168) but either "overstepped" or "crossed". In places, it is even impossible to understand what the authors are trying to say: What, e.g., does he following mean? "Sudden-onset events may reflect only short-term areas of the regions' threats..." (p. 16, ll. 553-554). I really have no idea...

Author Response

1) Introduction has been broadened by key relevant references on the links between climate change, migration and conflict;   2) Research design has been developed: we concentrated on Afghanistan and removed all discussions regarding other regions in order to present specific case of the links between climate change, migration and conflict. The developed research design as well as methods have been described in the introduction in detail.   3) The results and conclusions have been developed in accordance with new research design.   4) We will order proofreading services to improve the language level of our paper 

Reviewer 2 Report

The scope of the research seems very broad wide. In a single paper, it has attempted to cover a broad range of issues. Rather focusing on a particular region or an issue could help ensure novelty.  No concrete methodology is provided for the study. It is important to know for the readers how the research is conducted. For example, one of the aims of research is to identify "hotspot" and regions and drivers of displacement in the regions. But no methodology is provided how the aim will be achieved, through empirical research or desktop based study.      While the lit review is significantly extensive drawing on from other published materials, the analysis/discussion part is not fully aligned, and lack originality.  The author has included a number of figures 1-4, table 2 and the source of these is author himself. But no specific reference/evidence is provided in this regard.   

Author Response

1) Introduction has been broadened by key relevant references on the links between climate change, migration and conflict;   2) Research design has been developed: we concentrated on Afghanistan and removed all discussions regarding other regions in order to present specific case of the links between climate change, migration and conflict. The developed research design as well as methods have been described in the introduction in detail.   3) The results and conclusions have been developed in accordance with new research design.   4) We will order proofreading services to improve the language level of our paper.   

Reviewer 3 Report

This is a most useful synthesis of the environmental displacement discussion. Disaggregating slow and rapid onset events, as well as trying to tease out their international geographies is a useful framework. Adding the urbanization dimension is also useful as much of the discussion focuses on rural displacements and ignores urban vulnerabilities. If revisions are required a clearer concluding comment linking the point about law specifically to the details in the framework might help, but its not a necessary condition for acceptance of the paper. A careful grammar check will help in a few places, such as line 185, but this too isn't a condition for paper acceptance.

Author Response

Thank you very much for your positive feedback. We will order proofreading services to improve the language level of our paper.  

Round 2

Reviewer 2 Report

The revised article is now rather focused on nexus between climate change-displacement and conflict, particularly in Afghanistan. However, the title of the article is not aligned with the contents of the article. So the author is requested to revise the title.

Author Response

Thank you very much for your comment. We ' ve revised the title. Our new title is: Nexus between climate change, displacement and conflict: Afghanistan case
